# Predictive Factors of Early and One-Year Mortality in Patients with Acute Pancreatitis

**DOI:** 10.3390/diagnostics16010116

**Published:** 2026-01-01

**Authors:** Ana Sekulic, Olivera Marinkovic, Novica Nikolic, Milica Brajkovic, Barbara Loboda, Teodora Aleksijevic, Jasna Gacic, Igor Nadj, Stefan Guslarevic, Danilo Milic, Sladjana Trpkovic, Aleksandar Pavlovic, Darko Zdravkovic

**Affiliations:** 1University Hospital Medical Center Bežanijska Kosa, 11000 Belgrade, Serbia; olivera.marinkovic67@gmail.com (O.M.); novica.nikolic87@yahoo.com (N.N.); brajkovic.milica@yahoo.com (M.B.); barbara.loboda@hotmail.com (B.L.); aleksijevic.teodora@gmail.com (T.A.); jasna.gacic37@gmail.com (J.G.); 90igor90@gmail.com (I.N.); drdarkozdravkovic@gmail.com (D.Z.); 2Faculty of Medicine, University of Belgrade, 11000 Belgrade, Serbia; 3Institute for Orthopaedic Surgery Banjica, 11000 Belgrade, Serbia; danilo.milic.99@gmail.com; 4Medical Faculty, University of Pristina, 38220 Kosovska Mitrovica, Serbia; sladjana.trpkovic@gmail.com (S.T.); sasaaleksandarpavlovic@gmail.com (A.P.)

**Keywords:** acute pancreatitis, mortality, prognosis, scoring systems, sepsis, intensive care unit (ICU)

## Abstract

**Background/Objectives:** Acute Pancreatitis (AP) is an unpredictable inflammatory disease associated with high morbidity and significant mortality, particularly in severe forms. Early death is primarily linked to Systemic Inflammatory Response Syndrome (SIRS) and Multi-Organ Failure (MOF). The objective of this study was to identify objective clinical and laboratory predictors of early and one-year mortality in AP patients and to evaluate the prognostic accuracy of commonly used severity scoring systems. **Methods**: This prospective, observational study enrolled 50 adult patients admitted to the Intensive Care Unit (ICU) at the University Hospital Center Bežaniska Kosa. Patients with chronic pancreatitis, trauma-induced AP, or late presentation were excluded. Severity scores (APACHE II, BISAP, Ranson, Pancreas) and biomarkers (C-reactive protein, Procalcitonin) were collected at admission (0 h) and dynamically at 48 h, 72 h and day 7. Endpoints were early (in-hospital) and one-year mortality. **Results**: Overall mortality was 16% (*n* = 8). Mortality was significantly associated with sepsis/septic shock (*p* < 0.001), severe AP (*p* = 0.001), prolonged mechanical ventilation, and ICU stay. At admission, APACHE II (AUROC 0.813) and BISAP (AUROC 0.807) showed good accuracy. Reassessment at 48 h markedly improved prediction: APACHE II achieved excellent value (AUROC 0.917), and the Ranson score became a strong predictor (*p* < 0.001). Procalcitonin (PCT) was identified as a significant and superior predictor of mortality from 48 h onwards (*p* < 0.001), outperforming CRP. One-year survival was significantly shorter among patients with sepsis, septic shock, severe AP, and prolonged ICU stay. **Conclusions**: Dynamic assessment using clinical scoring systems, particularly APACHE II and BISAP within the first 48 h, provides reliable mortality prediction in acute pancreatitis. The presence of sepsis, severe disease, and the need for prolonged organ support are key mortality determinants. Serial PCT monitoring offers sensitive, incremental value for risk stratification and guiding intensive care decisions in both short- and long-term outcomes.

## 1. Introduction

Acute pancreatitis (AP) is an acute inflammatory disease of the pancreas that may result in local pancreatic injury, systemic inflammatory response, and organ failure [1]. It represents one of the most common gastrointestinal emergencies with an unpredictable clinical course and uncertain outcomes [2].

The global incidence of AP ranges from 4.9 to 73.4 cases per 100,000 population, with a rising trend worldwide [3,4]. The incidence varies across Europe: it is lower in England, the Netherlands, and Croatia; moderate in Germany and Scotland; and highest in Finland and Poland, largely related to high alcohol consumption [5,6,7,8].

Overall mortality in AP is about 5%, but rises to 10–30% in severe forms, and in some studies, reaches up to 50% [9,10]. The most frequent causes are gallstone disease and alcohol abuse, followed by hyperlipidemia, post-endoscopic retrograde cholangiopancreatography (ERCP), pancreatic neoplasms, certain drugs, metabolic disorders, hereditary syndromes, and idiopathic forms [1,11,12].

The diagnosis of AP is established if at least two of the following three requirements have been met: abdominal pain, three-fold increase in the pancreatic amylase and/or lipase values with respect to the upper reference limit and a positive finding on the contrast-enhanced computed tomography (CT) or, magnetic resonance (MR) or transabdominal ultrasonography (US) [4]. Upon diagnosis establishment, it is very important to assess the severity of the disease and the treatment outcome. According to the Atlanta classification from 2012, there are three grades of this disease: mild, moderate and severe AP [8,13,14,15,16].

Early mortality is most often due to systemic inflammatory response syndrome (SIRS) and multiple organ failure (MOF), while late mortality is related to sepsis from infected pancreatic necrosis [17].

Given the heterogeneity of AP and the variability in host inflammatory response, mortality prediction represents a major challenge, especially in studies with small sample sizes. Therefore, findings derived from limited cohorts should be interpreted cautiously, as low event rates substantially reduce statistical power and may overestimate the predictive performance of severity scores and biomarkers.

The aim of this study was to identify simple and objective predictors of early and one-year mortality in patients with AP and to evaluate the discriminative ability of commonly used scoring systems.

## 2. Materials and Methods

This was a prospective observational study conducted at the ICU University Hospital Center Bežanijska Kosa, Belgrade. The research was conducted with the approval of the Ethics Committee and management of the University Hospital Center Bezanijska Kosa which is responsible for educational, scientific and research activities. Number 3442/3 date 10 May 2018.

### 2.1. Inclusion Criteria

Adult patients (>18 years);Both sexes;Confirmed diagnosis of AP (≥2 of the following): (1) typical abdominal pain; (2) amylase/lipase ≥ 3 × upper limit of normal; (3) imaging findings consistent with AP [4,5].

### 2.2. Exclusion Criteria

Pregnant patients;Presence of chronic pancreatitis;Occurrence of AP following a multiple trauma episode;Patients who were admitted to the University Hospital Center Bežanijska Kosa from another healthcare facility, and more than 48 h had passed since the diagnosis of acute pneumonia (AP);Having history of organ transplant.

### 2.3. Data Collection

Demographic and clinical data;Etiology of AP;Laboratory parameters (C reactive protein—CRP, Procalcitonin—PCT, routine biochemistry);Imaging (US, CT, MRI);Severity scores: Ranson, Pancreas Score, Acute Physiology and Chronic Health Evaluation II score—APACHE II and Bedside Index for Severity in Acute Pancreatitis—BISAP.

### 2.4. Study Protocol

Scores were calculated at admission (day 0), at 48–72 h, and on day 7. Immediately after hospital admission, the following clinical scoring systems were calculated: Ranson score, Pancreas score, APACHE II score, and BISAP score. Concurrently, laboratory parameters including CRP and PCT were measured.

At 48 h after admission, CRP and PCT levels were reassessed, and all scoring systems (Ranson score, Pancreas score, APACHE II score, and BISAP score) were recalculated.

At 72 h and 7 days after admission, laboratory investigations were repeated (including CRP and PCT). At these time points, only the BISAP score was calculated among the scoring systems.

Outcomes were assessed at discharge and at 12 months. Each patient or a family member was contacted to determine vital status (alive or deceased). In cases of death, the date of death was recorded.

### 2.5. Endpoints

Early mortality (in-hospital);One-year mortality;Predictive accuracy of scoring systems (Area Under the Receiver Operating Characteristic Curve—AUROC analysis).

Systemic Inflammatory Response Syndrome (SIRS) was defined according to the established consensus criteria as the presence of at least two of the following: body temperature > 38 °C or <36 °C; heart rate > 90 beats/min; respiratory rate > 20 breaths/min or arterial partial pressure of carbon dioxide (PaCO_2_) < 32 mmHg; and white blood cell count > 12 × 10^9^/L, <4 × 10^9^/L, or >10% immature (band) forms. The previous definition of sepsis, based on the presence of suspected or confirmed infection accompanied by a systemic inflammatory response, has been replaced by the Sepsis-3 definition.

Sepsis was defined according to the Third International Consensus Definitions for Sepsis and Septic Shock (Sepsis-3) as a life-threatening organ dysfunction caused by a dysregulated host response to infection. Organ dysfunction was operationalized as an acute increase in the Sequential Organ Failure Assessment (SOFA) score of ≥2 points from baseline in the presence of suspected or confirmed infection. In patients without documented pre-existing organ dysfunction, the baseline SOFA score was assumed to be zero. Suspected or confirmed infection was determined based on clinical evaluation, microbiological findings, and/or radiological evidence, in accordance with routine ICU practice.

Multi-organ failure (MOF) was defined as concurrent dysfunction of two or more organ systems, quantified using the SOFA score. MOF was considered present when dysfunction involved at least two organ systems, each contributing to an elevated SOFA score, reflecting clinically significant impairment of respiratory, cardiovascular, renal, hepatic, neurological, and/or hematological function. The SOFA score was assessed at predefined time points in parallel with severity scoring to ensure consistent evaluation of organ dysfunction over time [18].

### 2.6. Statistical Analysis

Student’s *t*-test, Chi-square, Mann–Whitney, Kruskal–Wallis, ANOVA, Spearman rank correlation, and Cox regression were applied. Significance was set at *p* < 0.05.

Due to the limited sample size (*n* = 50) and particularly low number of outcome events (*n* = 8), all statistical analyses were considered exploratory. Regression models, AUROC analyses, and survival curves were interpreted with caution because small event numbers increase the risk of model instability, wide confidence intervals, and potential overfitting. Multivariable modeling was not performed or was restricted to avoid violating the rule of events per variable.

## 3. Results

A total of 50 patients were included (52% male, 48% female). The overall mortality rate was 16%. Because only eight deaths occurred in the cohort, all mortality associations reported below should be interpreted cautiously as exploratory findings rather than definitive conclusions. No significant differences were observed between survivors and non-survivors regarding sex, pancreatic necrosis, or total length of hospitalization. Mortality was significantly associated with: presence of sepsis/septic shock (*p* < 0.001), longer duration of mechanical ventilation (*p* < 0.001), prolonged ICU stay (*p* = 0.018) and severe AP (*p* = 0.001) (Table 1).

Although mortality showed statistical associations with sepsis/septic shock, severe AP, prolonged mechanical ventilation, and ICU stay, these relationships may be unstable due to the very small number of events and should be interpreted as exploratory.

The present analysis in Table 2. demonstrates that clinical scoring systems, particularly APACHE II and Ranson scores, provide the most reliable prediction of outcome in acute pancreatitis. At admission, APACHE II and BISAP scores showed good discriminative ability, with AUROC values above 0.80, confirming their utility as early prognostic tools. 

APACHE II and BISAP scores demonstrated signals of prognostic discrimination at admission; however, AUROC estimates obtained in small samples with few events may overestimate true predictive ability.

In contrast, serum biomarkers such as CRP and PCT demonstrated only moderate accuracy, with procalcitonin performing poorly at baseline.

Reassessment at 48 h markedly improved prognostic accuracy. The APACHE II score achieved excellent predictive value (AUROC 0.917) with both high sensitivity and specificity. 

APACHE II reached a high AUROC estimate at 48 h, although such high values should be interpreted cautiously because small event numbers increase the risk of model overfitting. Ranson score also showed strong performance (AUROC 0.856). Although BISAP maintained acceptable accuracy, it was less powerful than APACHE II and Ranson at this time point. Among biomarkers, CRP showed only modest accuracy, whereas PCT achieved very high sensitivity but lacked specificity, making it unsuitable as a standalone marker. 

PCT showed a statistical association with mortality from 48 h onward; however, the apparent magnitude of this association may be inflated by limited statistical power and should be viewed as preliminary.

BISAP demonstrated stable and moderate discriminative ability across all later time points (AUROC 72 h 0.780 and AUROC 7 d 0.783. In contrast, CRP and PCT showed variable and generally limited prognostic value, especially at later stages.

These findings are consistent with previous studies reporting that dynamic scoring systems, which incorporate evolving physiological and clinical data, outperform static biochemical markers in predicting severity and mortality in acute pancreatitis. In clinical practice, early use of APACHE II or BISAP at admission may help in risk stratification, but repeated evaluation at 48 h with APACHE II or Ranson scores provides the most reliable prognostic information for guiding intensive management strategies. Among the evaluated scoring systems, BISAP showed the most consistent performance across all time points; however, given the limited number of outcome events, these findings should be regarded as exploratory.

In Table 3, Scores of 3–4 on Ranson at admission were significantly associated with mortality (*p* = 0.002). Scores >5 at 48 h were highly predictive of death (*p* = 0.0012). Pancreas scores between 3 and 8 at both time points correlated significantly with mortality (*p* = 0.009 and *p* = 0.004). Higher APACHE II scores (20–29) at baseline and 48 h predicted mortality. These associations should not be interpreted as independent predictive effects, as the number of deaths was insufficient to support stable multivariable modeling. BISAP scores ≥3 at admission and ≥5 at 48 h, 72 h, and 7 days showed strong correlation with mortality, especially at 48 h and 7 days (*p* = 0.002–<0.001). Although BISAP showed consistent statistical associations with mortality across multiple time points, these results reflect exploratory signals rather than confirmed predictive validity.

Dynamic evaluation of severity scores, particularly APACHE II and BISAP during the initial 48 h, provides reliable prediction of mortality, aiding clinical decision-making.

Dynamic evaluation of severity scores, particularly APACHE II and BISAP during the initial 48 h, suggested improved prognostic patterns over time, but reliability cannot be assumed due to limited event counts.

Figure 1a shows that the CRP values did not differ statistically significantly between patients with a fatal outcome and patients who survived. In patients who survived, there was a constant sharp increase in CRP values, which reached a maximum after 72 h and then a decline was recorded. In patients with a fatal outcome, there were smaller oscillations between measurements, but the values were significantly higher compared to CRP values in patients who survived. The change in CRP values was not statistically significantly different in the observed time in both groups of patients.

Figure 1b shows that the PCT values were statistically significantly higher in patients with a fatal outcome compared to patients who survived. Also, PCT values differed statistically significantly over the observed time. Namely, over the examined time there was a statistically significant linear trend of PCT value growth, without significant differences in PCT values between individual measurements. Changes in PCT values did not differ statistically significantly over the observed time in both groups of patients.

Because biomarker-based hazard ratios derived from small datasets are prone to wide confidence intervals, the precision of these associations is limited.

Survival time was statistically significantly shorter in patients who had sepsis or septic shock as a complication (*p* = 0.023) (Figure 2), who had severe AP (*p* = 0.002) (Figure 3) and who were obese (*p* < 0.001) (Figure 4).

These findings represent statistical trends within this cohort; however, they may not be generalizable and require validation.

Table 4 presents the results of the Cox regression analysis for different severity scoring systems in acute pancreatitis. The BISAP score demonstrated a consistent and statistically significant association with mortality at all time points (0 h, 48 h, 72 h, 7 d). Each 1-point increase being associated with an almost twofold increase in the risk of death (HR 1.70–1.95, *p* < 0.01). 

Hazard ratio estimates suggested increased mortality risk with higher BISAP values, though these effect sizes may be unstable due to sparse events.

The Ranson score was not a significant predictor at admission (*p* = 0.078), but became a strong and independent predictor of mortality after 48 h (HR 2.21, *p* < 0.001).

The APACHE score was already significant at admission (HR 1.22, *p* = 0.001), and its predictive strength further increased at 48 h (HR 1.36, *p* < 0.001).

The Pancreas score showed a significant association with mortality at admission (HR 1.84, *p* = 0.017), while this association was no longer statistically significant at 48 h (*p* = 0.105).

These results should not be interpreted as evidence of independent predictors, as event numbers were insufficient to support robust regression.

Table 5 presents the results of the Cox regression analysis for CRP, PCT, duration of mechanical ventilation, ICU stay, and total length of hospitalization.

CRP levels during the first 72 h were not significant predictors of mortality, whereas after 7 days a statistically significant but modest association with poor outcome was observed. In contrast, PCT emerged as a strong and consistent predictor of mortality from 48 h onwards, with higher values significantly increasing the risk of death. That means that PCT has greater prognostic value than CRP in predicting mortality among patients with acute pancreatitis, particularly after 48 h of hospitalization.

Among clinical parameters, prolonged mechanical ventilation and longer ICU stay were identified as independent predictors of mortality. In contrast, the overall length of hospitalization showed no statistically significant association with survival.

Although these variables demonstrated an association with mortality in unadjusted analyses, the limited number of events precluded assessment of their potential independent prognostic value.

## 4. Discussion

In this study, we investigated predictive factors of early and one-year mortality in patients with acute pancreatitis (AP). Acute pancreatitis (AP) remains a clinically heterogeneous disease, ranging from mild localized inflammation to severe forms involving multiple organ systems [19]. Approximately 15 to 25 percent of all patients with acute pancreatitis (AP) develop moderately severe or severe AP. Although 95% to 98% of patients survive the disease, it is still one of the leading causes of death among patients hospitalized with gastrointestinal disorders in the United States and Europe [20]. About 50% of deaths occur within the first 2 weeks of the disease and early death is usually associated with persistent multiple organ failure (MOF) [10,21]. Organ failure in AP is associated with up to 30% mortality rates [22].

In our study, overall mortality was 16%, with the majority of deaths occurring in patients with severe AP and those who developed complications such as sepsis or septic shock. Most patients with a fatal outcome (62.5%) had a severe form of AP. Pancreatic necrosis was present in 48% of patients compared to the total number of patients studied and 50% compared to the number of patients with a severe form of AP, but this was not statistically significant in relation to the outcome of treatment. Complications such as sepsis and septic shock were statistically significantly more common in patients with a fatal outcome compared to the survivors.

However, given the very small number of deaths (*n* = 8), these associations may be unstable and should be interpreted cautiously as exploratory rather than confirmatory.

Our findings confirm that systemic complications, particularly sepsis, septic shock, and multiple organ failure (MOF), remain the main determinants of early mortality, whereas infection of necrotic pancreatic tissue is strongly associated with late mortality. These observations are consistent with prior studies, which emphasize that mortality in AP follows a biphasic pattern: early deaths due to overwhelming systemic inflammatory response and organ dysfunction, and late deaths linked to septic complications [9,17,22].

Consistent with previous literature, gallstones and alcohol abuse were the leading etiologies of AP in our cohort, although the relative distribution differed slightly between sexes. Hyperlipidemia, increasingly reported in Asian populations, was identified as the predominant cause in large Chinese cohorts [23]. Overweight and obesity were frequent in our population, with 47.6% of patients having a body mass index above the normal range. Importantly, obese patients demonstrated significantly shorter one-year survival, suggesting that excess body weight may contribute to more severe disease and worse long-term outcomes [24].

These findings should be regarded as preliminary due to the limited sample size and low event rate.

Several clinical scoring systems are widely used to predict severity and mortality in AP. In our cohort BISAP score consistently predicted mortality at all time points (0 h, 48 h, 72 h, 7 d), with each point increase nearly doubling the risk of death. This aligns with evidence demonstrating its comparable or superior accuracy relative to APACHE II and Ranson scores [25,26]. Although BISAP demonstrated consistent associations with mortality, the small number of events likely inflates effect estimates, and these results should be viewed as hypothesis-generating.

APACHE II showed high predictive value already at admission, improving at 48 h, consistent with its established role in identifying patients requiring intensive care and early resuscitation [27]. The apparent strength of APACHE II in this analysis should be interpreted with caution because AUROC values derived from small samples are prone to overestimation.

Ranson score was not predictive at admission but became a reliable predictor after 48 h, reflecting its calculation requirements and delayed application [28].

Pancreas score demonstrated significant prognostic value at admission but lost statistical significance after 48 h, suggesting limited utility for longer-term risk stratification [29]. Due to sample size limitations, the observed loss of statistical significance at 48 h may reflect insufficient power rather than true absence of association.

These results underscore that different scoring systems have optimal predictive performance at different time points during hospitalization. Early use of BISAP and APACHE II can guide immediate triage decisions, whereas Ranson and Pancreas scores may provide more relevant information after the first 48 h. Nevertheless, because our sample included only 50 patients and 8 deaths, the relative performance of the scoring systems in this study should not be interpreted as definitive but rather as exploratory.

Analysis of one-year survival revealed that the APACHE II and Ranson scoring system are a statistically significant predictor of mortality. The degree of calibration of the Pancreas_0_ scoring system in terms of treatment outcome was quite high, immediately after the BISAP_0_ score and proved to be a significant predictor of one-year survival. These trends should be interpreted cautiously, as long-term analyses with few events (*n* = 8) are highly susceptible to model instability and overfitting.

C-reactive protein (CRP) and procalcitonin (PCT) were evaluated as adjunctive biomarkers. In our study, CRP at day 7 was a significant predictor of one-year mortality, while PCT values at 48 h, 72 h, and day 7 strongly correlated with mortality. However, biomarker-based hazard ratios must be interpreted conservatively due to sparse events and wide confidence intervals. Previous studies report that CRP ≥ 150 mg/L at 48 h distinguishes severe from mild AP with good sensitivity and specificity [28], while PCT is particularly sensitive for detecting early organ dysfunction and multi-organ failure [30,31]. Although initial CRP measurements may better predict mortality than initial PCT, monitoring the dynamic trends of both biomarkers offers additional prognostic information, with CRP reflecting disease severity and PCT providing early sensitivity to systemic complications [32]. However, it has also been proved that in order to predict the progression of AP the initial PCT values were less accurate than the Ranson and BISAP scores, which showed significantly better correlation [33]. Most guidelines on AP advise against the use of a single marker to triage patients [28]. In our research, after 48 h, both inflammatory biomarkers had lower prognostic value for treatment outcome compared to scoring systems [34,35]. We have proven in our research that PCT, in combination with clinical indicators of treatment intensity, may serve as a reliable marker for risk stratification and clinical decision-making. These associations require validation, as small numbers of critically ill patients in this cohort may have disproportionately influenced the observed results.

Therefore, while clinical scoring systems are most useful at the time of admission, biomarkers may provide incremental value during the early hospital course, particularly in guiding decisions about antibiotic therapy and invasive interventions [17].

Our study also confirmed that prolonged need for mechanical ventilation and extended intensive care unit (ICU) stay are strongly associated with mortality. This finding underscores the importance of timely escalation of care, as patients requiring prolonged organ support represent the subgroup at highest risk of death. Previous studies have shown that the presence of persistent organ failure beyond 48 h is the single strongest predictor of mortality [10,13,35].

One-year survival was significantly shorter among patients with sepsis, septic shock, severe AP, and prolonged ICU stay. Although this analysis is limited in our study by the small number of long-term events.

These findings are consistent with prior reports highlighting age, comorbidities, diabetes, elevated CRP on admission, pleural effusions, pancreatic necrosis, and systemic complications as independent risk factors for reduced long-term survival [36]. Importantly, etiology influenced short-term outcomes but appeared less relevant for long-term survival. Some studies have found that higher scores on the scoring system (Ransonov, Pancreas, BISAP) and longer ICU stay may be predictors of mortality and long-term survival in univariate but not multivariate analyses [36,37].

### 4.1. Clinical Implications

Our findings support a multimodal approach for risk stratification in AP, integrating clinical scoring systems with inflammatory biomarkers. Early identification of high-risk patients can guide intensive monitoring, prompt intervention, and appropriate allocation of resources. While scoring systems such as BISAP and APACHE II provide robust early prediction, Ranson and Pancreas scores remain useful for ongoing assessment, and serial CRP and PCT measurements offer additional insights into disease progression [25,26,27]. Yet, given the exploratory nature of our findings, these tools should not be used to alter clinical practice based solely on this study.

### 4.2. Limitations

The primary limitation of this study is the small sample size (*n* = 50) and particularly the low number of mortality events (*n* = 8), which substantially reduces statistical power. Under these conditions, AUROC analyses, survival curves, and Cox regression models are prone to instability, wide confidence intervals, and overfitting. Therefore, all findings should be interpreted as exploratory. Additional limitations include single-center design, limited generalizability, and inability to perform robust multivariable analyses.

Larger, multicenter studies are warranted to validate these findings and explore the utility of emerging machine learning models, which have demonstrated superior predictive accuracy in recent investigations [38].

## 5. Conclusions

Dynamic assessment using APACHE II and BISAP within the first 48 h provides reliable mortality prediction in acute pancreatitis. Ranson and Pancreas scores appeared to provide complementary prognostic information. Serial PCT measurement is a sensitive early predictor, whereas CRP trends contribute to the evaluation of disease progression. Prolonged mechanical ventilation, longer ICU stay, and sepsis or septic shock were linked to poorer outcomes. These observations should be interpreted with caution and regarded as preliminary signals rather than confirmatory evidence. Integrating these tools allows for effective risk stratification, targeted intervention, and improved patient management in both short- and long-term outcomes [38]. However, due to the small overall sample and very low number of deaths, these associations may overestimate the true predictive performance of the evaluated tools.

Overall, the findings of this study should be viewed as exploratory and hypothesis-generating. Larger multicenter studies with adequate numbers of events are required before firm conclusions can be drawn or applied to clinical decision-making.

## Figures and Tables

**Figure 1 diagnostics-16-00116-f001:**
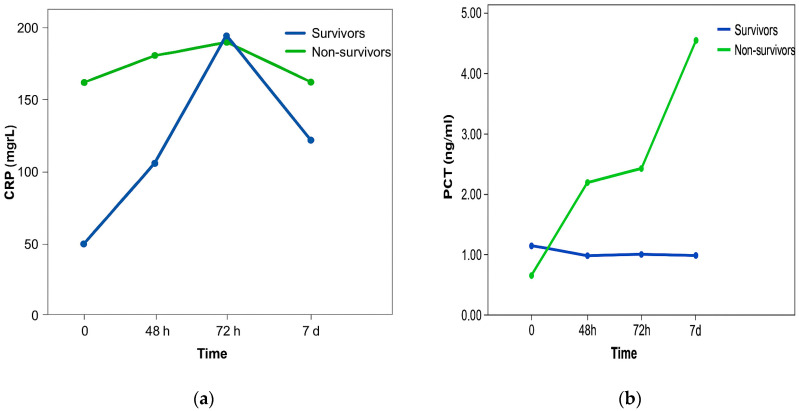
CRP (**a**) and PCT (**b**) values between patients with fatal outcomes and patients who survived, and the relationship of CRP values over the observed time.

**Figure 2 diagnostics-16-00116-f002:**
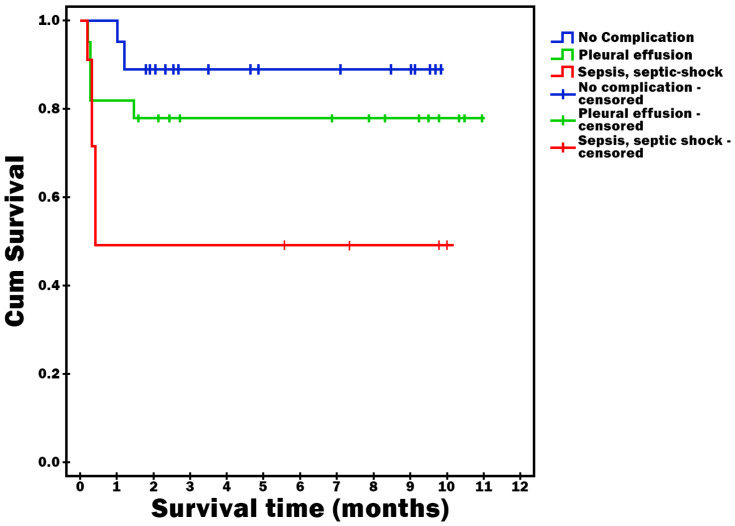
Survival time vs. Complications.

**Figure 3 diagnostics-16-00116-f003:**
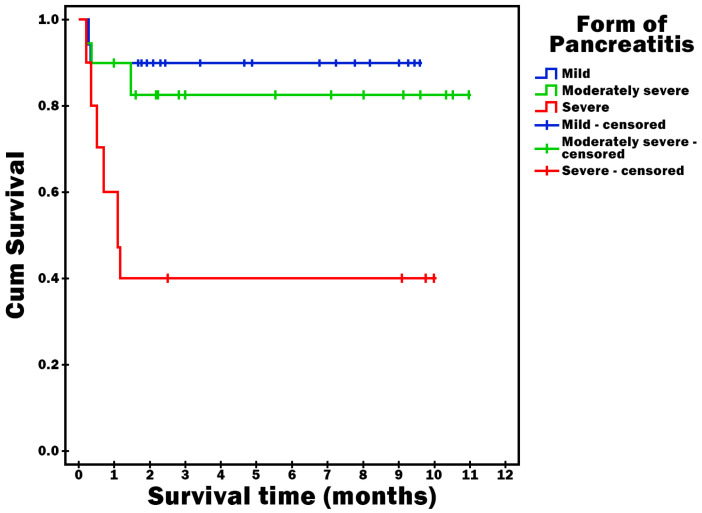
Survival time vs. Form of AP.

**Figure 4 diagnostics-16-00116-f004:**
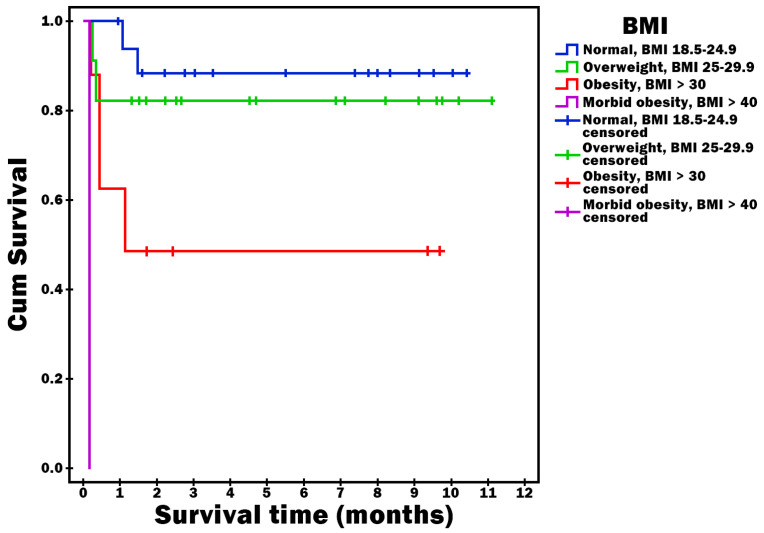
Survival time vs. BMI.

**Table 1 diagnostics-16-00116-t001:** Demographic and clinical characteristics of patients with acute pancreatitis in relation to outcome.

Characteristic	Total *n* = 50 (%)	Survivors *n* = 42 (84%)	Non-Survivors *n* = 8 (16%)	*p*
Sex				
Male	26 (52.0)	22 (52.4)	4 (50.0)	0.902
Female	24 (48.0)	20 (47.6)	4 (50.0)	
Etiology				
Gallstones	30 (60.0)	26 (61.9)	4 (50.0)	
Hyperlipidemia	8 (16.0)	6 (14.3)	2 (25.0)	
Alcohol abuse	4 (8.0)	4 (9.5)	0 (0)	
Unknown	8 (16.0)	6 (14.3)	2 (25.0)	
Necrosis				0.902
Present	24 (48.0)	20 (47.6)	4 (50.0)	
Absent	26 (52.0)	22 (52.4)	4 (50.0)	
Complications				<0.001 *
Without complications	22 (44.0)	22 (52.4)	0 (0)	
Pleural effusion	18 (36.0)	16 (38.1)	2 (25.0)	
Sepsis/septic shock	10 (20.0)	4 (9.5)	6 (75.0)	
BMI				
Normal weight	17 (40.5)	17 (40.5)	0 (0)	
Overweight	24 (47.6)	20 (47.6)	4 (50.0)	
Obesity	8 (16.0)	5 (11.9)	3 (37.5)	
Morbid obesity	1 (2.0)	0 (0)	1 (12.5)	
Duration of MV (days), median (range)	0 (0–20)	0 (0–2)	4.5 (0–20)	<0.001 *
Duration of ICU stay (days), median (range)	2 (0–20)	1.5 (0–20)	3.0 (0–3)	0.018 *
Length of hospitalization (days), median (range)	15 (6–62)	12.5 (6–62)	17.5 (9–27)	0.183
Form of pancreatitis				0.001 *
Mild	20 (40.0)	18 (42.9)	2 (25.0)	
Moderately severe	20 (40.0)	19 (45.2)	1 (12.5)	
Severe	10 (20.0)	5 (11.9)	5 (62.5)	

* *p* < 0.05 indicates statistically significant. Abbreviations: BMI—body mass index; MV—mechanical ventilation; ICU—intensive care unit.

**Table 2 diagnostics-16-00116-t002:** Predictive value of severity scores and biomarkers in acute pancreatitis.

Parameter	AUROC	95% CI	Cut-Off	Sensitivity (%)	Specificity (%)
Day 0 (at admission)					
Ranson (0 h)	0.693	0.547–0.816	≤2.0	59.5	87.5
APACHE II (0 h)	0.813	0.677–0.909	≤15.0	69.0	87.5
Pancreas (0 h)	0.695	0.549–0.817	≤2.0	50.0	87.5
BISAP (0 h)	0.807	0.670–0.905	≤2.0	69.0	87.5
CRP (0 h)	0.753	0.611–0.864	≤46.3	64.3	87.5
PCT (0 h)	0.580	0.432–0.718	≤0.3	66.7	75.0
48 h after admission					
Ranson (48 h)	0.856	0.727–0.939	≤5.0	92.9	87.5
APACHE II (48 h)	0.917	0.803–0.976	≤17.0	90.5	87.5
Pancreas (48 h)	0.729	0.585–0.845	≤2.0	57.1	87.5
BISAP (48 h)	0.789	0.650–0.891	≤2.0	69.0	87.5
CRP (48 h)	0.667	0.519–0.794	≤183.2	90.5	50.0
PCT (48 h)	0.545	0.398–0.686	≤2.1	97.6	37.5
72 h after admission					
BISAP (72 h)	0.780	0.640–0.885	≤2.0	78.6	87.5
CRP (72 h)	0.506	0.361–0.650	≤202.0	64.3	12.5
PCT (72 h)	0.673	0.525–0.799	≤0.5	69.1	75.0
7 days after admission					
BISAP (7 d)	0.783	0.637–0.891	≤3.0	92.1	62.5
CRP (7 d)	0.513	0.361–0.663	≤281.0	92.1	37.5
PCT (7 d)	0.709	0.556–0.833	≤2.0	92.1	62.5

Abbreviations: AUROC—Area Under the Receiver Operating Characteristic curve; Day 0—measurement at admission; 48 h—measurement 48 h after admission; 72 h—measurement 48 h after admission; 7 d—measurement 7 days after admission; APACHE II—Acute Physiology and Chronic Health Evaluation II score; BISAP—Bedside Index for Severity in Acute Pancreatitis; CRP—C-reactive protein; PCT—procalcitonin.

**Table 3 diagnostics-16-00116-t003:** Association of severity scores with final outcome in acute pancreatitis.

Score/Time	Category	Total *N* (%)	Survivors *N* (%)	Non-Survivors *n* (%)	*p*
Ranson (0 h)	3–4	25 (50.0)	17 (40.5)	8 (100.0)	0.002 *
Ranson (48 h)	3–4	16 (32.0)	16 (38.1)	0 (0)	
	>5	21 (42.0)	13 (31.0)	8 (100.0)	0.0012 *
Pancreas (0 h)	3–8 (high risk for severe AP)	29 (58.0)	21 (50.0)	8 (100.0)	0.009 *
Pancreas (48 h)	3–8	27 (54.0)	19 (45.2)	8 (100.0)	0.004 *
APACHE II (0 h)	10–19	32 (64.0)	28 (66.7)	4 (50.0)	
	20–29	10 (20.0)	6 (14.3)	4 (50.0)	0.021 *
APACHE II (48 h)	10–19	32 (64.0)	30 (71.4)	2 (25.0)	
	20–29	10 (20.0)	4 (9.5)	6 (75.0)	<0.001 *
BISAP (0 h)	3–4	17 (34.0)	11 (26.2)	6 (75.0)	0.008 *
	5	4 (8.0)	2 (4.8)	2 (25.0)	
BISAP (48 h)	3–4	11 (22.0)	9 (21.4)	2 (25.0)	
	5	10 (20.0)	4 (9.5)	6 (75.0)	0.002 *
BISAP (72 h)	3–4	8 (16.0)	4 (9.5)	4 (50.0)	
	5	8 (16.0)	4 (9.5)	4 (50.0)	
BISAP (7 d)	3–4	8 (17.4)	4 (10.5)	2 (25.0)	
	5	8 (17.4)	4 (10.5)	4 (50.0)	<0.001 *

* *p* < 0.05 indicates statistically significant association with mortality. Abbreviations: APACHE II—Acute Physiology and Chronic Health Evaluation II; BISAP—Bedside Index for Severity in Acute Pancreatitis; AP—acute pancreatitis.

**Table 4 diagnostics-16-00116-t004:** Survival analysis—Cox regression model of BISAP, Ranson, APACHE, and Pancreas scoring systems.

Scoring System	Beta	*p*-Value
BISAP 0 h	0.67	0.003 *
BISAP 48 h	0.59	0.005 *
BISAP 72 h	0.53	0.006 *
BISAP 7 d	0.60	0.004 *
Ranson 0 h	0.52	0.078
Ranson 48 h	0.79	<0.001 *
APACHE 0 h	0.20	0.001 *
APACHE 48 h	0.31	<0.001 *
Pancreas 0 h	0.61	0.017 *
Pancreas 48 h	0.36	0.105

* Statistically significant (*p* < 0.05). Abbreviations: BISAP—Bedside Index for Severity in Acute Pancreatitis; AP—acute pancreatitis; APACHE—Acute Physiology and Chronic Health Evaluation.

**Table 5 diagnostics-16-00116-t005:** Survival analysis—Cox regression model of CRP, PCT, duration of mechanical ventilation, ICU stay, and length of hospitalization.

Parameter	Beta	*p*-Value
CRP 0 h	0.004	0.072
CRP 48 h	0.004	0.296
CRP 72 h	0.003	0.208
CRP 7 d	0.006	0.022 *
PCT 0 h	−0.084	0.667
PCT 48 h	0.757	<0.001 *
PCT 72 h	0.686	<0.001 *
PCT 7 d	0.260	<0.001 *
Duration of MV	0.138	0.001 *
Duration of ICU stay	0.132	0.006 *
Length of hospitalization	−0.003	0.894

* Statistically significant (*p* < 0.05). Abbreviations: CRP—C-reactive protein; PCT—Procalcitonin; MV—Mechanical Ventilation; ICU—Intensive Care Unit.

## Data Availability

The data that support the findings of this study are available from the corresponding author (A.S.) upon reasonable request. The data are not publicly available due to privacy and ethical restrictions.

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
