# Peer review of "Diagnostics2026, 16(1), 116;https://doi.org/10.3390/diagnostics16010116"

_diagnostics, 2026, doi:10.3390/diagnostics16010116_

Round 1

Reviewer 1 Report

Comments and Suggestions for Authors

1) The authors should describe in the materials and methods how sepsis, Systemic Inflammatory Response Syndrome (SIRS) and Multi-Organ Failure (MOF) were defined.

2) The authors state that all severity scores and biomarkers of inflammation were measured on admission (0h), at 48h, 72h and on day 7. They also state that the end points of the study were in-hospital (early) and one-year mortality. Nevertheless, the results do not show data on all time points. In addition, there are no results on MEWS. Furthermore, it is not clear at which time point the reported deaths refer. These points need elucidation and the results should be more clearly presented and commented.

3) Please give abbreviations in full when first mentioned in the text, see PCT, BISAP, MEWS, AUROC.

4) Please pay attention to some grammatical errors: ($p=0.001$) in the Abstract, 48–72 h should be 48h, 72h, Legend of Figure 3, inour (line 400), avoid using bullets in the discussion and make clear that the values in parentheses in the second column of Table 1 represent percentages.

5) Please avoid excessive repetition though the manuscript of the limitation statements regarding your results.

Reviewer 2 Report

Comments and Suggestions for Authors

The article presents an interesting work of patients with severe acute pancreatitis who were managed in the ICU

department. But there are some issues to be addressed:

Although, there were only 8 death, at 25% of them (2 patients) the etiology is not known. Due to the fact that alcohol abuse produce severe pancreatitis with high mortality in younger patients, how the author could explain the fact that alcohol abuse does not produce any death I'm their cohort?
